# Integration of Finite Element Analysis and Laboratory Analysis on 3D Models for Methodology Calibration

**DOI:** 10.3390/s24134048

**Published:** 2024-06-21

**Authors:** Sara Gonizzi Barsanti, Rosa De Finis, Riccardo Nobile

**Affiliations:** 1Department of Engineering, University of Campania Luigi Vanvitelli, 81100 Caserta, Italy; 2Department of Engineering of Innovation, University of Salento, 73100 Lecce, Italy; rosa.definis@unisalento.it (R.D.F.); riccardo.nobile@unisalento.it (R.N.)

**Keywords:** 3D models, NDT sensors, integration of sensors, calibration, structural analysis, FEA

## Abstract

To better address mechanical behavior, it is necessary to make use of modern tools through which it is possible to run predictions, simulate scenarios, and optimize decisions. sources integration. This will increase the capability of detecting material modifications that forerun damage and/or to forecast the stage in the future when very likely fatigue is initiating and propagating cracks. Early warning outcomes obtained by the synergetic implementation of NDE-based protocols for studying mechanical and fatigue and fracture behavior will enhance the preparedness toward economically sustainable future damage control scenarios. Specifically, these early warning outcomes will be developed in the form of retopologized models to be used coupled with FEA. This paper presents the first stage of calibration and the combination of a system of different sensors (photogrammetry, laser scanning and strain gages) for the creation of volumetric models suitable for the prediction of failure of FEA software. The test objects were two components of car suspension to which strain gauges were attached to measure its deformation under cyclic loading. The calibration of the methodology was carried out using models obtained from photogrammetry and experimental strain gauge measurements.

## 1. Introduction

To realize studies to identify the level of deterioration of objects, it is fundamental to have proper, correct and accurate documentation to arrange restoration and reinforcement interventions. This is necessary prior to the selection of preservation methods and materials. To analyse the structural behavior of an object, finite element analysis is usually employed. This is a recognized method in engineering for the modelling analysis of stress behavior. Normally, the most used pipeline implies the use of 3D models built with Non-Uniform Rational B-spline (NURBS) surfaces. This is the most common way to represent the ideal shape of the object that is going to be simulated [1]. The passage from a mesh model, made of superficial surfaces, to a volumetric one, can be carried out through the creation of NURBS and then the export of these poly-surfaces as step files. The volumes needed in FEA software Ansys R19.2, in fact, differ from a surface mesh typically generated with 3D capture methods in the nodes distributed both on the exterior surface and the interior volume connected to each other by elementary volumes such as tetrahedrons, pyramids, prisms or hexahedral.

This workflow is suitable in the mechanical discipline because usually the physical object analysed and simulated is like its model drawing within rigorous tolerances.

Currently, the documentation provided by a 3D survey with active sensors such as laser scanning and passive technologies such as photogrammetry has been widely developed.

The models that can be obtained from these techniques are precise and accurate, but their purpose is different from that of FEA. These models, as said, are shaped by high-resolution exterior triangular surfaces and hence are represented by millions of polygons. This geometry is not appropriate for the direct use of these models in FEA software Ansys 19.2. Hence, the conversion of these models into volumes is mandatory. The density of the element composing the mesh must also be reduced from millions to thousands because the computational complexity of FEA grows exponentially with the number of nodes that are inside the volumetric model representing the objects that will be simulated. Since the level of simplification needed is strong, one possibility is to use the retopology process. This process allows us to strongly reduce the number of elements on a mesh, also adding a topological rearrangement of the surface. This also means the creation of a new topology for the 3D model [2] made of quadrangular instead of triangular elements. The use of this kind of polygonal element allows us to obtain a mesh with a better distribution of the element on its surface. This new rearrangement of elements and the quadrangular shape allows, then, a substantial reduction in the number of the polygons that compose the retopologized model. In addition, the proposed process takes into account the suitability of the simplified mesh to be converted into a set of NURBS surfaces through an automatic procedure. This allows the production of volumetric models of an object or structure maximizing the closeness of the resulting NURBS model with the acquired one, and in the meantime minimizing the number of NURBS patches required for describing it. Since NURBS are made of quadrangular patches, it is suggested that a more organized topology of the initial mesh model could be advantageous for the conversion of the polygonal mesh into NURBS. The homogeneity of shapes between retopology models and NURBS is supposed to allow a better coherence of shapes and geometry with the digitized artefact. The methodology proposed in this paper is based on an intelligent use of retopology processes combined with a conversion of these simplified and retopologized models to mathematical NURBS surfaces as closely as possible to the real shape of the object surveyed. These final models are supposed to be suitable for the transformation in rationally multifaceted volumetric models through standard FEM packages.

The focus of this work is to present a first stage of calibration of the different sensors for the correct use of 3D reality-based models for structural analysis, integrating the data from active and passive sensors with laboratory and computational data. The validation of the methodology is fundamental to identifying the correct level of simplification of 3D reality-based models to be used in FEA software Ansys 19.2 to obtain an accurate result. This process will be calibrated on two parts of suspensions as a representative of real components because they can be easily tested in the laboratory. In particular, the strains acquired during laboratory tests were successfully compared with those obtained by FEA analysis.

Future work will apply this methodology in those fields where the complex geometry and the impossibility of carrying out laboratory tests require the development of new methodologies, i.e., the cultural heritage field, integrating the survey with thermographic data [3]. The test of the process with mechanical parts that can be tested in the laboratory can validate the methodology and calibrate the system. It is known that the application of this methodology on objects much bigger and with more complex geometry can imply a level of approximation that is higher than the error calculated in these tests. The process is nevertheless important to give a basis for further investigations.

The integration of data acquired using different sensors will improve the analysis and the interpretation of the results, addressing issues such as incompleteness or distortion, so the combined use of 3D survey and thermography will help to identify potential damages in a more in-depth way. Specifically, the idea will be to improve the process automation by integrating surveyed 3D models and IRT in the phase of data acquisition and data modelling together with the improvement of the Field of View (FoV) of each IRT image to provide a three-dimensional reading of the thermal behavior of the object [4]. The advantage will lie in the combination of different types of information, including accurate measurements, thermal data, and different reactions of deteriorated materials in the near-infrared spectrum geometric data to prevent damage and plan future interventions effectively.

### 1.1. Three-Dimensional Reality-Based Survey

Three-dimensional techniques for the digitization of objects, buildings, and sites involve imaging sensors [5] mounted on the satellite, aerial and unmanned aerial vehicles/Remotely Piloted Aircraft System (UAV/RPAS) platforms or hand-held devices and ranging sensors like laser scanners or structured light instruments [6]. These two approaches are image-based and range-based modelling (IBM and RBM, respectively). Often, both approaches are integrated to leverage the intrinsic advantages of each one and overcome potential problems such as low accuracy in the scaling of photogrammetric models and missing parts resulting from problems during the acquisition phase.

Photogrammetry is “the process of deriving (usually) metric information about an object through measurement made on photographs of the object” [7]. Active systems, especially those based on laser light, operate regardless of the light and texture of the object being detected because they modify the exterior appearance using suitably coded light. The modelling is carried out through a series of three-dimensional coordinates, usually embedded in a reference system that has its origin in the centre of the instrument. All scanning systems operate through an almost completely automatic process through which they can acquire many points per second, even reaching up to one million.

### 1.2. Mesh: Triangular-Based and Quad-Dominant

Topology is mentioned as the study of geometrical characteristics and three-dimensional relations between the polygonal elements composing the superficial part of a mesh. This goes independently by constant deviation of shape and size of these elements, while any unexpected modification in this relationship is a topological error. The most common errors are as follows: (i) flip of the normal in two adjacent polygons; (ii) self-intersecting polygons; (iii) non-manifold polygons; (iv) zero area faces; (v) duplicated faces, etc. Triangular meshes are described by a string of triangular elements, the barycentre of which defines a linear surface representation. A mesh can be labelled as a series of vertices.
V = {v1, …, vV}(1)
and a series of triangles bonding them
F = {f1, …, fF}, fi ∈ V × V × V(2)
even if is more efficient to define the triangular mesh with the edges of the polygons
ε = {e1, …, eE}, ei ∈ V × V(3)
since the connectivity occurs on the triangle’s edges. In summary, “each edge stores references to its endpoint vertices, to its two incident faces and to the next and previous edge within the left and right face, respectively, while vertices and faces store a reference to one of its incident edges” [8].

The reconstruction of superficial meshes starting from an unorganized oriented 3D point cloud is relatively difficult, mainly because the sampling of the point is frequently non-uniform and because of the noise of the cloud. This is due usually to the fact that the normals are often not coherent among each other due to sampling imprecision and misregistration of the different scans.

The creation of a mesh starts, therefore, from an unorganized point cloud and needs to produce an organized structure made of triangles. Hence, the process assumes the topology of the unidentified surface, trying to adjust in a more coherent and fitted way the noisy data and filling holes realistically following the surface reconstruction. It is, of course, clear that if the 3D point cloud is strongly noisy, the interpolation among points for the creation of the mesh results in a wrong topology, with several errors and incorrect data. These data need to be corrected because they can lead to a wrong simulation in the FEA software Ansys 19.2. There are different methods to create a mesh from a point cloud, for example, the Delaunay triangulations, the alpha shapes, or the Voronoi diagrams that interpolate all or most of the points. With noisy data, the resultant surface is often rough, smoothed or refit to the points in subsequent processing. Other methods, such as global or local approaches, directly reconstruct an approximating surface. Global fitting methods define the implicit function as the sum of radial basis functions (RBFs) centred at the points. Local fitting methods consider subsets of nearby points at a time. A simple scheme is to estimate tangent planes and define the implicit function as the signed distance to the tangent plane of the closest point. Finally, Poisson reconstruction creates very smooth surfaces that robustly approximate noisy data. Poisson systems are well known for their resilience in the presence of imperfect data [9].

Topology based on quadrangular elements is basically applied mostly in 3D modelling for videogames, CG animation or artistic standards, primarily because these models are created from scratch and are basically composed of rows and columns, which allows creating a simpler, cleaner, and more structured mesh that can be manipulated and modified in an easier way, for example, if a subdivision in different parts is needed. The use of this process also allows for increased edge loops and is much more appropriate if the models need to be changed in shape, deformed, or segmented for animation purposes. The topology of a quad-based model is simpler than the triangular one and permits adjustment of edge flow easily. The outcome, in this way, can be straightforwardly subdivided. A triangle-based model can lead to the creation of artefacts on the topology of the model, like sharp angles and edges that can alter the design of the surface. With quadratic elements, on the other hand, edge loops can be easily added or manipulated to obtain a smoother deformation of the surface; furthermore, quad elements follow the geometry of the surface in a better way than triangular ones due to their shape, hence the possibility of strongly simplifying the mesh without losing geometric description and accuracy. The method based on retopology, therefore, subdivides the triangular high-resolution mesh at a three-dimensional resolution that is lower than the initial one maintaining a degree of accuracy higher than the one that can be obtained by sampling the mesh with triangular elements. This is because retopology preserves the overall geometry of the initial mesh, describing from zero its topological structure. It is called quadrangle mesh because is mainly made by quads except for some unavoidable triangular polygons [10]. Retopology is a key 3D process that has been developed for optimizing the use of polygons in describing 3D shapes in Computer Graphics (CG) animations, where the rendering time is directly related to the number of polygons of the represented 3D geometry [11]. Retopology is the creation of a new topology for a 3D model. In practical applications, like computer animation, it is obtained by laying down a low-polygon mesh over the top of the high-density model to simplify it and in the meantime start a brand-new polygonal organization, possibly created to follow the main geometrical feature of the object described by the 3D model. The retopologized mesh is typically based on quadrangular elements (quads) instead of triangles. In this way, animators can rework such models with shapes close enough to the original but without the huge number of polygons typical of models originated by a 3D digitization pipeline. This also simplifies all the processing stages following the modelling, like the rigging, or the action needed for transforming a static 3D model of a character into a dynamic entity, capable of moving, walking and running.

### 1.3. Non-uniform Rational B-Spline—NURBS Curves and Surfaces

NURBS is the abbreviation for Non-Uniform Rational B-Spline and can be defined as follows: non-uniform refers to the parametrization of the curve and allows for the occurrence of multi-knots needed to characterize Bézier curves; rational expresses the mathematical representation, which allows NURBS to represent accurately conic surfaces. A uniform parametrization represents that integral quantities are set to the points so that the first one will always be 0.0, the second one 1.0, the third one 2.0, and so on. The value of the parameter of the last point will always be the number of lengths of the curve. NURBS curves are a simplification of a formulation, represented in the form of a ratio between two polynomial expressions, that make them “rational”. A NURBS curve is defined by its order, a set of weighted control points, and a knot vector. Control points regulate the shape of the curve while a knot vector is a system of parameter values defining where and how the control points modify the NURBS curve. A NURBS surface (patch) is obtained by a series of NURBS curves in two directions (called “U” and “V”) interpolated to create a surface as the tensor product of two NURBS curves, originating a quadrangular patch. NURBS surfaces have the same advantages as their generating curves, namely that they can correctly represent conical shapes, they have a local control, and they allow the realization of complex shapes without elevating the degree of polynomials excessively. Starting from the supposition that a mesh represents 3D surfaces as a series of discreet faces that are made of polygons and that the smoothness depends on the number of polygons defined when reconstructing a mesh, as much as pixels represent an image, the NURBS surfaces are mathematical representations of curves and surfaces that are made of parametric cubic curves, so edges are connected to form a continuous grid-like surface and can represent complex forms. A NURBS-based 3D model is, in general, the composition of a certain number of NURBS patches (NURBS surfaces) connected to maintain a positional (G0), tangential (G1), or curvature (G2) continuity [12]. Patches are two-parameter equivalents of curve segments defined using blending functions in two independent parameters and a set of control points. For the generation of curve segments, linear, quadratic, and cubic interpolation methods are used; therefore, bilinear, biquadratic, and bi-cubic polynomial interpolation methods are adopted for constructing surface patches. CAD software as Rhinoceros allows us to directly convert a polygonal model into a NURBS model. This, in general, tends to create a higher number of small patches (usually a number close to the one of elements that shape the mesh) when the original mesh is topologically unorganized. On the contrary, when dealing with mesh composed of quadrangular elements, the conversion led to a four-sided untrimmed 1-degree NURBS surface. This means that the edges of the surface are the same as the ones representing the mesh face’s edges. Triangular meshes will be converted into either trimmed or untrimmed planar NURBS surfaces so that the resulting polysurface will have the same edges as the original mesh model and be composed entirely of 1 × 1 degree (bilinear) NURBS surfaces. By rearranging the initial topology of the mesh, a preliminary condition for minimizing the number of NURBS patches of the converted model is set, and this represents a better starting point for the mesh tool embedded in the standard FEM package.

### 1.4. Finite Element Analysis

The analysis of the structural behavior of an object can be afforded using finite element analysis, a well-established technique in engineering for modelling stress performances. The finite element method (FEM) is a mathematical method used to accomplish finite element analysis (FEA). This process was primarily developed and used for structural behavior in the mechanic field and then used for answering other categories of problems that can vary from dynamic to fluid, electrical, thermal, and biomaterial. The physical problem that has to be solved typically contains a structure or structural component dependent on specific loads. It is, hence, fundamental to idealize these problems into mathematical ones, involving precise conventions that can be dealt with differential equations. The finite element analysis resolves this mathematical model, and because this result is a numerical process, it is crucial to take into consideration the accuracy of the solution: if this accuracy is not reached, the numerical result must be reiterated with a refinement of the parameters, for example, using a finer mesh, and hence different types or different dimensions of elements, until a satisfactory accuracy is achieved. Commonly, mathematical models are used in FEA as an idealization of the physical one. This idealized model is created to predict or simulate the behavior of the physical object considered in the analysis. These models, meshed with specific elements, are then used for carrying this out. The choice of the elements to be used in the meshing module of the FEA software Ansys 19.2 depends on the type of model imported and the type of problem to be solved. The elements differ from 2D or 3D: for 2D models, triangular and quadrangular elements can be used, depending on the structure of the model. Quad elements are usually preferred over triangular ones because they provide a higher accuracy in the results. In general, 3D elements are tetrahedral and hexahedral and, as for the 2D elements, the latter ones provide more accurate results because the deformation is lower in the strain energy state. On the contrary, the meshing process with hexahedral elements is more difficult because the number of nodes is higher, so a segmentation of the model is usually needed, especially if the geometry of the object is complex [13]. The 3D elements can be subdivided into linear or quadratic elements. The main difference is that with quadratic elements, nodes are present also in the mid side, in a number that varies from 4 nodes (linear tetrahedron) to 20 nodes (quadratic hexahedron). Studies have been carried out to analyse the behavior of the different types of elements used in the meshing process. To summarize, Ref. [14] give these suggestions:(1)Do not use linear tetrahedron elements because they are too rigid.(2)Quadratic tetrahedral elements can always be used.(3)Linear hexahedral elements are sensible with respect to the corner angle, so large angles in stress concentration regions must be avoided.(4)Quadratic hexahedral elements are very robust but computationally expensive.(5)For thin-walled structures, the limit of element edge/thickness ratio to use tetrahedra is about 2000.

Finite element analysis approximates the solution of the problem submitted to the analysis and the behavior of each point internal to the finite element is labelled through nodal displacement. This is the first result of FEM computation. This displacement is defined by the combination of nodal movement through the shape functions, which explains the displacement inside a component. The formulation of this is contingent on the accuracy of the result. As the deformations are interconnected to the derivative of the displacement and the stresses to the deformations across the material, the accuracy of these features is correlated to the form functions.

The integration and comparison of FEA and laboratory tests are well established in the literature, starting from the medical field [15,16], experimental data using a digital image correlation engine [17,18] and to determine the tensile loading effects and displacement [19]. Integration and comparison have also been conducted to calculate the effect of specimens’ design and manufacturing process on micro tensile bond strength and specimens’ integrity [20] and the simulation of performances of different models of buried pipes have been examined through comparative study [21]. The innovative part of the presented work is the direct use of 3D reality-based models in the FEA software Ansys 19.2 instead of the custom process of drawing the part from scratch.

## 2. Materials and Methods

The methodology was tested and calibrated on two components of car suspension positioned inside the frame of the car (Figure 1a,b). They have been coloured to better identify the parts in the structure. The two parts that have been analysed are the red and the green ones: the differences between them are the length and the fact that the red part has three small holes along the central part.

### 2.1. Experimental Campaign

Calibrating a component involves measuring the strain using strain gauges after applying a load and comparing it with the theoretical strain to obtain a mechanical characterization of the component. Electrical resistance strain gauges (SG) measure the change in electrical resistance of a conductor when subjected to deformation. The conductor is a cohesive grid with the strain gauge applied to the specimen surface. The strain ε is given by the relation
(4)$$\varepsilon =\left(1/K\right)\cdot \left(∆R/R_{0}\right)$$
where K is the SG calibration factor, R0 is the initial resistance, and ΔR is the resistance variation in the SG. The gage factor was 2.155 ± 0.5%.

This variation is measured using the Wheatstone bridge. The Wheatstone half-bridge connection [22,23,24,25,26] was used for all suspension components to compensate for spurious deformation (Figure 2a–f).

The installation of strain gauges (SG) followed technical standards UNI 10478-3: ‘Strain gauge installation and verification’.

Strain gauges were applied to both the intrados and extrados of the components (Figure 2a,b), with grids mounted in both longitudinal and transverse directions. The strain gauges BE6 and AE5 were mounted on the red component (Figure 2a), while the strain gauges AE3 and BE4 were mounted on the green component (Figure 2b). The experimental data were acquired using the digital acquisition system provided by HBM^®^ Quantum X composed of modules CX27, MX1615, and MX840A.

The loading system was designed to apply a unidirectional tensile/compressive stress using the component bushing holes (Figure 2c,d). The loading machine used was an MTS 810 with a capacity of 100 kN. The applied loads were typical forces that load the suspension components, as suggested by the car producer. The tests were conducted using load control mode, applying a force ranging from 0 to 3000 N, followed by a return to 0 N, and inducing a compression from 0N to −3000 N. The cycle concluded with unloading.

The use of strain gauges made it possible to evaluate the strain state of the components and trace the calibration curves. Finally, knowing the displacements of the extremities of the components where the load was applied, using displacement transducers on the loading machine, an estimation of the stiffness of the components in the loading direction was evaluated.

### 2.2. Survey and Creation of Volumes for FEA

The objects (Figure 3a,b) have been surveyed both with photogrammetry and laser scanning to compare and integrate the data. For the photogrammetric survey, a Canon 5D(MAnufacturer: Canon Owned by University Of Campania Vanvitelli, Aversa, Italy) coupled with a 24 mm lens was used (Figure 4a). The parameters of the camera have been set to ISO 400 and aperture to 4. This was obliged by the environmental condition of the room where the survey was performed: it was necessary to set an external lighting to improve the illumination of the objects. Fortunately, even if metallic, the material was compliant with the photogrammetric process being opaque and with a good texture. The models have been processed with Agisoft Metashape and scaled using a metal scale and specific metric targets (Figure 3b).

The scanner used was a FARO Edge ScanArm CAM2 (Manufacturer: FARO Owned by University of Salento, Lecce, Italy) (Figure 4b) that combines a portable CMM (Coordinate Measuring Machine) with a 3D scanner, allowing it to capture intricate details and dimensions, up to 560.000 points per sec. It has an accuracy of ±25 μm, a 7-axis repeatability of 0.029 mm, a 7-axis accuracy of ±0.041 mm and the area of acquisition is 80 × 150 mm. The instrument has a camera that detects the distortion of the laser in the object that is translated as variations in height, hence defining the superficial profile and the position of the object, taking into consideration the position of its own base. The triangulation is possible using three sizes: distance between the source of the laser and the camera; the angle of the emitter of the laser; and the angle of the camera.

The acquisitions were made using the Geomagic Control X software. The results of the survey consisted of 3D meshes directly aligned during the survey. The result of the survey with this active device was not completely appreciable because the results presented anomalies on the surfaces, probably due to an improper calibration of the instrument (Figure 5). In the future, the same scanner, calibrated, will be used along with other scanners more compliant with the surface acquired. This will be carried out to identify the best survey technique for this type of artefact and to analyse if the modelling errors can be avoided using laser scanning techniques.

It was decided to use these models, cleaned, as references to increase the accuracy of the scaling of the photogrammetric models and to integrate them if some parts were missing.

The 3D models have been oriented following the Cartesian axis displayed on Ansys, the software used for FEA, to apply in the most correct way the loads and the constrains, and then simplified by applying three different parameters—the mean *Y*-axis up, *X*-axis toward the right and *Z*-axis toward the operator.

The first simplification was a triangular one, conducted in Agisoft Metashape, imposing a fixed number of polygons on the mesh in 100K detail. The second and the third simplifications applied retopology using the open software InstantMeshes [14], setting the number of elements to half and then a quarter of the number of high-resolution models. NURBS have been then exported from these 3D models using Rhinoceros software and imported into Ansys for FEA. The details are listed in Table 1, and it is interesting to note that for the retopologized models, the number of patches of the resultant NURBS is almost half the number of the quadrangular elements, while for the triangular simplification, the number of patches is close to the number of triangles. This can be explained considering the shape of the elements: patches in NURBS are quadrangular and are organically disposed along the surface of the model, as with the elements in the retopologized models. Triangular elements are not compliant with the shape of the patches; plus, they follow the geometry of the surface less well while decreasing the number.

The models have been compared with the high-resolution reference to individuate the mean and standard deviation of each simplified mesh, also calculating their accuracy (Figure 6a–f). The details of the comparison are listed in Table 2.

### 2.3. Finite Element Analysis

The main part of the work is related to validating the geometric accuracy of the process, hence, to evaluate the results of the analysis carried out directly on 3D volumes derived from reality-based models in the FEM process on compression and tension of the two components. To certify it, the results of FEA have been compared to theoretically calculated strains and laboratory tests performed on the specimens.

For the structural analysis, the volumetric models have been imported into Ansys and then all the parameters have been set (Figure 7a,b):The material assigned was aluminium, for which a Young’s modulus of 68GPa and a Poisson ratio of 0.3 have been imposed in the data type section.The meshing module has used quadratic tetrahedral elements with a sizing put close to the size of the superficial element.A fixed support has been imposed on one side of the specimen all around the hole.A traction/compression force has been applied longitudinally to the length of the specimen along the *X*-axis.

Regarding the imposition of force and boundary conditions on the different models, it must be said that a complication arose while selecting the parts. As said, the common process uses CAD models made of continuous NURBS (Figure 8a,b). In this case, it is easy to select the parts when imposing the parameters. On the other hand, when the same process needs to be carried out on volumes created from reality-based models, the part is not composed of one single element but several small ones. While selecting, it is not difficult to choose the wrong elements, leading to a wrong imposition of forces or boundary conditions. To avoid this problem, once the NURBS was created in Rhinoceros, a Boolean difference was applied to create a proper correct and accurate hole, drawing a cylinder to cut the surface accordingly. In this way the hole is correct and so is the selection (Figure 9).

## 3. Results

The FEA was conducted on the three different models for each specimen, calculating the maximum principal elastic strain for traction in AE3/AE5 and for compression in BE4/BE6 and the minimum principal elastic strain for compression in AE3/AE5 and for traction in BE4/BE6. The volumes calculated from the triangular simplification of each specimen failed to mesh in Ansys. While the idea is to test the different simplifications using the same settings, it was decided not to try a different meshing procedure, for example, changing the dimension of the tetrahedral or linear elements instead of quadratic.

The red specimen gave better results with both the simplification processes, with a slight increase in the percentage of errors in the models with a stronger retopology simplification. This is probably because the shape of the specimen is geometrically simple without holes along the main body and because of its lower curvature than the green one. The green specimen, on the other hand, gave the worst results with the more simplified models. This is maybe because the holes in the main body were open in post-processing, while in the initial photogrammetric model, they were closed, adding uncertainty to the measurements (Table 3, Table 4, Table 5 and Table 6; Figure 10 and Figure 11). There is also the possibility that the orientation of the model is not perfect and that there are some problems in the modelling process. The reasons why these analyses gave this very bad result will be investigated in future works.

As said, the results are indeed promising since there are not huge differences, and probably, with a better survey or a better selection of elements, while imposing the parameter of load and boundary condition, it will be possible to have an improvement.

The maximum stress for traction and compression was also evaluated for the red (Figure 12a–d) and the green suspension (Figure 13a–d).

As it is possible to observe, the region with higher stress is located at the centre of the arm, where the strain gauges were installed. The figures below illustrate the stress states in the arm following a tensile and compressive load. The green/blue-coloured zone indicates a compressive stress state, while the red-coloured zone indicates a tensile stress state. In the case of a compressive load, the red and blue-coloured zones will be reversed from the previous tensile case.

## 4. Discussion and Future Work

The possibility to use reality-based models for FEA is a promising process. The validation and calibration of the process and sensors in the laboratory using specimens is fundamental to setting up the main parameters to be used in the process. Many difficulties have been encountered during the process, mainly in the post-processing part:The photogrammetric models were accurate, but the holes were closed, so they were opened by hand, adding some uncertainty to the measurement.Triangular simplification has been proved to be ineffective, starting from a lower accuracy of the surface (decreasing the number of elements can increase spikes and peaks where the geometry is more complex) to the difficulties in meshing the related volumes in the FEA software Ansys 19.2. The reason for these problems can be identified in the shape of the elements that are not compliant with both the patches of the NURBS and the tetrahedral elements, which leads to a number of patches that is too high for the software to deal with. This can be overcome by further decreasing the number of elements of the mesh, but it will also mean less accuracy.Retopology is a valid instrument and, when applied to models of objects with simple geometry, permits a straightforward process with few bottlenecks. When dealing with complex objects, it can increase problems in post-processing; the resultant models can show missing elements or parts and it is not always easy to come to a solution.

As said, the process is, however, interesting to be further investigated. The most pressing step is to identify the best practice to obtain the most accurate volume for FEA. The process here presented, although accurate, is long and sometimes complex. Each passage adds a level of uncertainty:From point cloud to triangular, a superficial mesh that approximates the 3D cloud.Post-processing of the mesh (closing holes, check topology).From a triangular to quadrangular mesh, especially adding smoothing and sometimes losing details.Closing holes and checking the topology.NURBS creation is straightforward, but the volume obtained is not made of simple surfaces but of usually a number of patches half the elements of the mesh.FEA is, per se, an approximation.

Since the last point cannot be avoided, it is possible to improve the other. One possible idea is to skip the first two points and create volumes directly on point clouds using voxels. As seen, the process to obtain a model for structural analysis implies a sort of approximation that has to be summed to the approximation of the meshing process from a sparse 3D point cloud and the approximation of the simplification of the mesh to create a volume, the main issue is to start with the most truthful data possible, which can guarantee the geometrical accuracy and the less loss of details possible.

The main problems while dealing with this process consist of the following:The way for obtaining a volume is not yet clearly defined and may greatly influence the result.The balance between geometric resolution and confidence level of the simulated results is often not compliant with the shape of a volume originated by a 3D acquisition process.

Voxelization is for sure faster and avoids all the post-processing of the mesh and the creation of NURBS, but the parameters have to be chosen wisely since a strong smoothing is often added to the model. This process seems, however, the most promising in terms of timesaving and accuracy since it avoids all the problems related to the different steps needed when dealing with a 3D mesh and its transformation.

Another process to be tested in the future could be the use of digital image correlation to measure the surface displacement and strain over the entire area of interest.

The versatility of this technique is so great that it can range from purely structural mechanical applications to others in completely different contexts, although the reason for the investigation is the same: to have an efficient and reliable model to study the decay of the elastic properties of the component/good. This is why the idea for future work is to apply this tested and validated methodology to Cultural Heritage objects that cannot be tested in a laboratory.

## Figures and Tables

**Figure 1 sensors-24-04048-f001:**
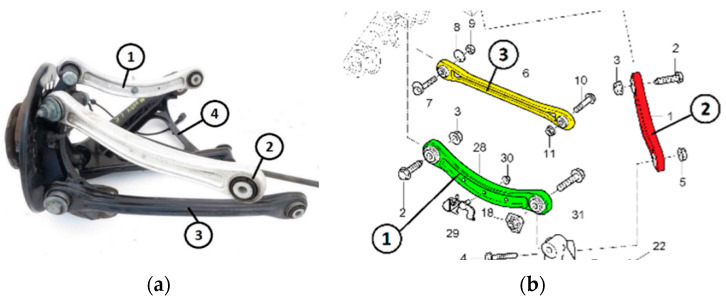
The position of the parts analysed in the car (**a**) and the specification with colours of the parts to better identify them (**b**).

**Figure 2 sensors-24-04048-f002:**
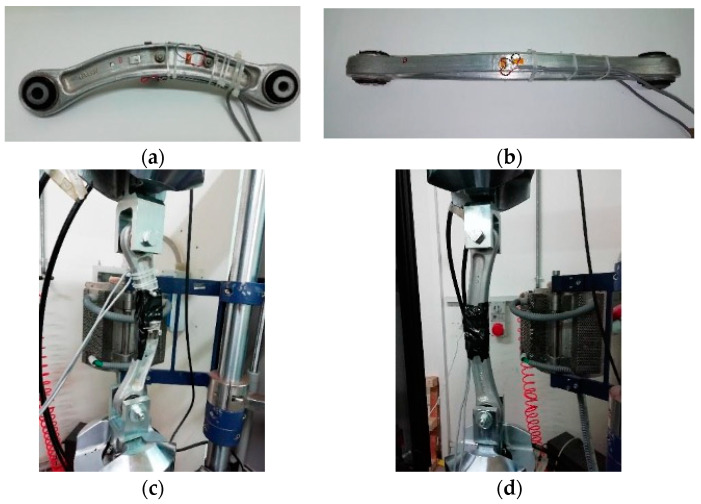
The installation of the sensors on the green specimen (**a**,**c**,**e**) and the red one (**b**,**d**,**f**).

**Figure 3 sensors-24-04048-f003:**
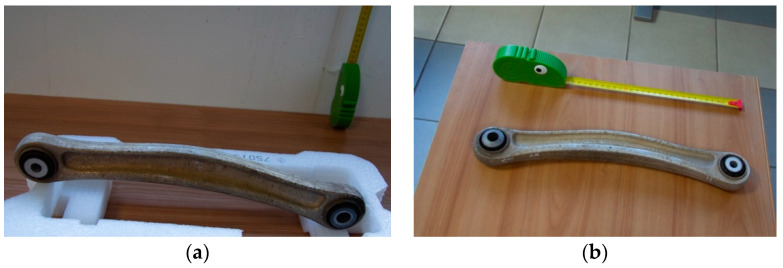
The two specimens, red (**a**) and green (**b**), with the scale bar used to scale them.

**Figure 4 sensors-24-04048-f004:**
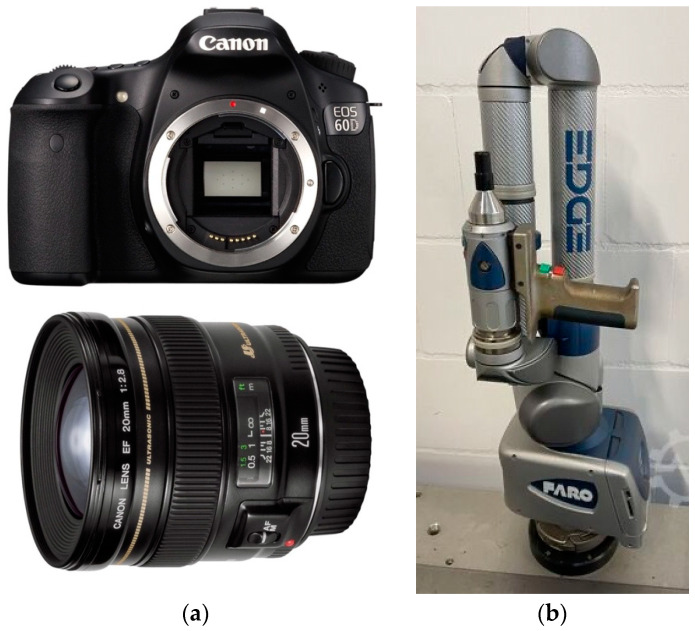
The (**a**) camera and lens and (**b**) Faro ScanArm2 used for the survey.

**Figure 5 sensors-24-04048-f005:**
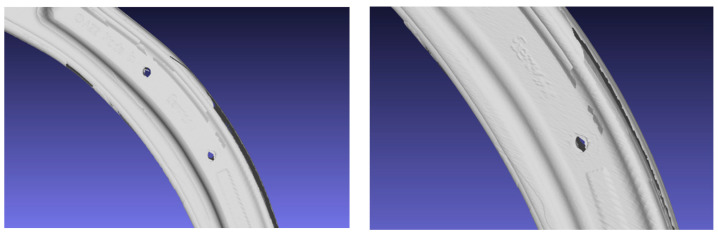
Errors and inaccuracy on the 3D models obtained with the scanner.

**Figure 6 sensors-24-04048-f006:**
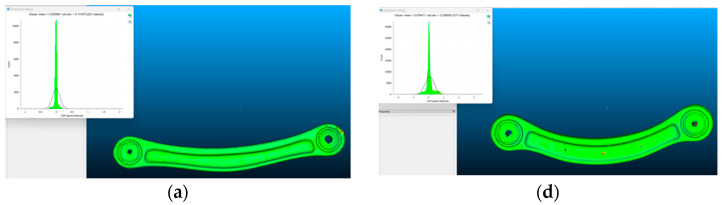
Comparison between the high-resolution photogrammetric models and the different simplifications. Red specimen: triangular (**a**), first medium simplification through retopology (**b**) and second, strong simplification through retopology (**c**). Green specimen: triangular (**d**), first medium simplification through retopology (**e**) and second, strong simplification through retopology (**f**).

**Figure 7 sensors-24-04048-f007:**
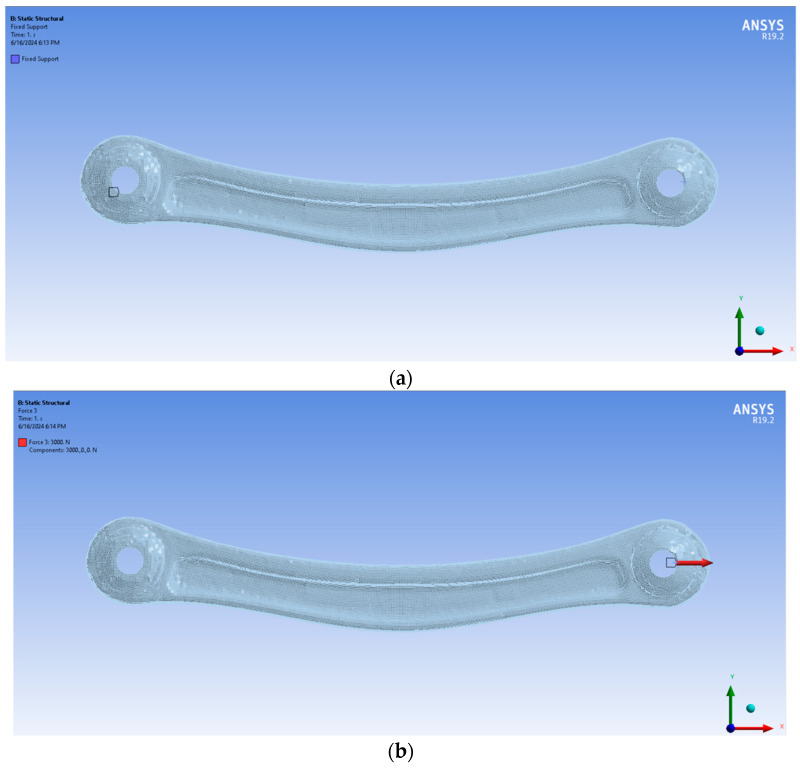
The setting of the fixed support on the entire hole (**a**) and the force along the *X*-axis (**b**).

**Figure 8 sensors-24-04048-f008:**
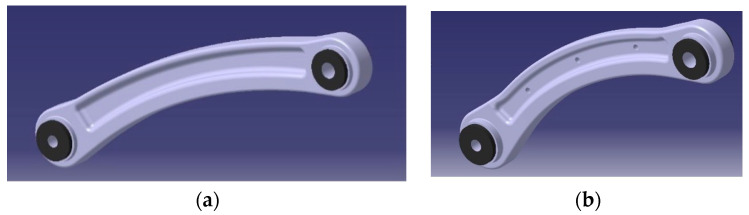
CAD models of the red (**a**) and green (**b**) specimens.

**Figure 9 sensors-24-04048-f009:**
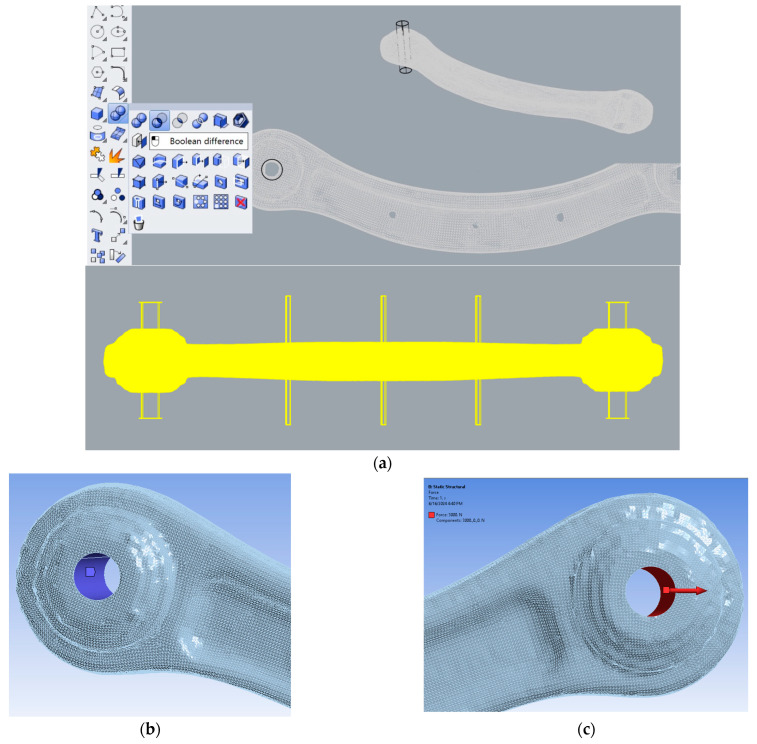
Boolean difference in Rhinoceros (**a**); selection of element on the green specimen for the definition of the fixed support highlighted in blue (**b**) and the traction highlighted in red (**c**).

**Figure 10 sensors-24-04048-f010:**
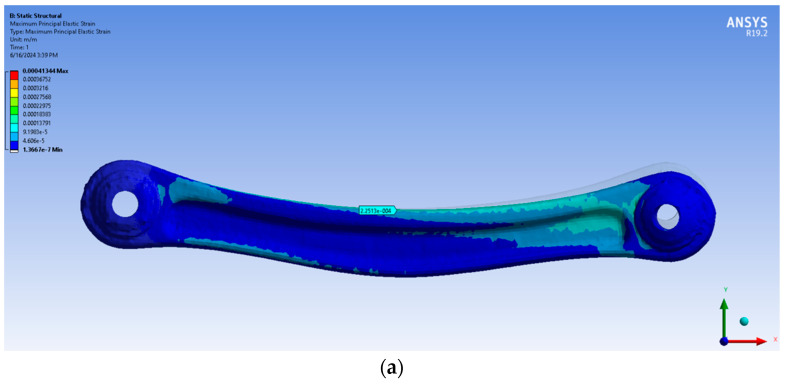
Results of FEA for the RED specimen: (**a**) traction in AE3; (**b**) compression in AE3; (**c**) traction BE4; (**d**) compression BE4 for the first retopology; (**e**) traction in AE3; (**f**) compression in AE3; (**g**) traction BE4; (**h**) compression BE4 for the second retopology.

**Figure 11 sensors-24-04048-f011:**
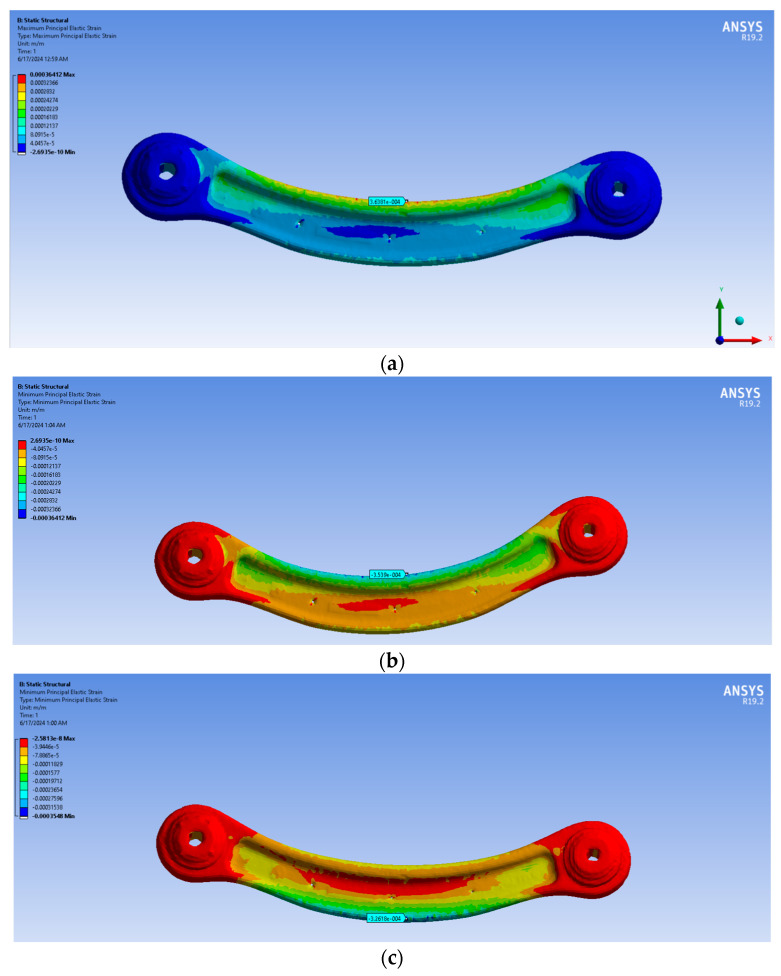
Results of FEA for the RED specimen: (**a**) traction in AE5; (**b**) compression in AE5; (**c**) traction BE6; (**d**) compression BE6 for the first retopology; (**e**) traction in AE5; (**f**) compression in AE5; (**g**) traction BE6; (**h**) compression BE6 for the second retopology.

**Figure 12 sensors-24-04048-f012:**
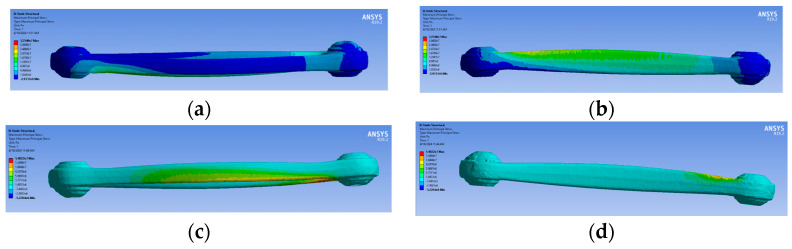
The maximum stress for traction (**a**) extrados (**b**) intrados (**c**) and compression, and for compression (**c**) extrados (**d**) intrados for red specimen second retopology.

**Figure 13 sensors-24-04048-f013:**
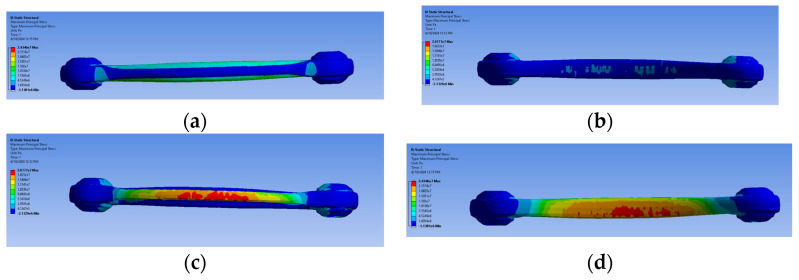
The maximum stress for traction (**a**) extrados (**b**) intrados (**c**) and compression, and for compression (**c**) extrados (**d**) intrados for green specimen second retopology.

**Table 1 sensors-24-04048-t001:** Number of elements and patches of the different models used.

	Green Specimen		Red Specimen	
	N° of Elements	NURBS	N° of Elements	NURBS
**Original file**	2,066,310	/	2,056,272	/
**Triangular mesh**	100,000	96,802	100,000	97,026
**1st retopology**	126,304	62,530	125,060	63,152
**2nd retopology**	31,540	15,394	30,788	15,770

**Table 2 sensors-24-04048-t002:** Mean and standard deviation in mm of the compared models.

	Green Specimen		Red Specimen	
	Mean	Standard Deviation	Mean	Standard Deviation
**Triangular mesh**	0.07643	0.2985	0.0027	0.1144
**1st retopology**	0.0013	0.0871	−0.0043	0.0783
**2nd retopology**	0.0607	0.2388	−0.0348	0.2234

**Table 3 sensors-24-04048-t003:** Results maximum and minimum strain in μm/m of FEA for the RED specimen first retopology.

	AE3 Traction	AE3 Compression	BE4 Traction	BE4 Compression
**Lab test**	228	−229	−124	124
**FEA**	225	−227	−121	123
**Error%**	−1.31	0.87	2.41	−0.81

**Table 4 sensors-24-04048-t004:** Results maximum and minimum strain in μm/m of FEA for the RED specimen second retopology.

	AE3 Traction	AE3 Compression	BE4 Traction	BE4 Compression
**Lab test**	228	−229	−124	124
**FEA**	227	−221	−120	122
**Error%**	−0.44	3.51	3.22	−1.61

**Table 5 sensors-24-04048-t005:** Results maximum and minimum strain in μm/m of FEA for the GREEN specimen first retopology.

	AE5 Traction	AE5 Compression	BE6 Traction	BE6 Compression
**Lab test**	489	−491	−345	347
**FEA**	363	−354	−326	342
**Error%**	−25.6	28	5.5	−1.44

**Table 6 sensors-24-04048-t006:** Results maximum and minimum strain in μm/m of FEA for the GREEN specimen second retopology.

	AE5 Traction	AE5 Compression	BE6 Traction	BE6 Compression
**Lab test**	489	−491	−345	347
**FEA**	353	−353	−302	293
**Error%**	−28	28.1	12.5	−15.6

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
