# Peer review of "Integration of Finite Element Analysis and Laboratory Analysis on 3D Models for Methodology Calibration"

_sensors, 2024, doi:10.3390/s24134048_

Round 1

Reviewer 1 Report

Comments and Suggestions for Authors

If the intent of the research is to perform characterization of Cultural Heritage objects as is stated at the end of the paper, it would be good to state that at the beginning of the paper and discuss limitations on measurements that can be performed on such objects.

Photogrammetry and laser scanning provide surface information. It does not appear that information was acquired on subsurface structures or flaws. How was this information acquired? 

A considerable amount of research has been done combining digital image correlation which can give very accurate in-plane and out of plane strain fields and finite element analysis. What are the relative advantages and disadvantages of the two techniques? 

A considerable amount of research has been done on generating FEM meshes from computed tomography point clouds. This was not mentioned in the paper nor was how these methodologies relate to developing FEM meshes from photogrammetry and laser scanning point clouds. It seems that some discussion of this would improve paper. 

It seems like there is too much detail on some well known areas such as how strain gauges work and mesh elements. If this is important for the audience, it seems a reference would be better. 

Comments on the Quality of English Language

English is understandable.

There are multiple typos in the paper such as: 

Line 55 The use of this king of  

Line 243 involving precise conventions that can be delt with differential  

Line 251 This models, meshed with specific elements, are then used for carrying out the. 

Line 253 the typo of model imported

Author Response

We thank the reviewer for their suggestions. We have answered each review below.

“If the intent of the research is to perform characterization of Cultural Heritage objects as is stated at the end of the paper, it would be good to state that at the beginning of the paper and discuss limitations on measurements that can be performed on such objects.”

We have added a sentence in the Introduction.

“Photogrammetry and laser scanning provide surface information. It does not appear that information was acquired on subsurface structures or flaws. How was this information acquired?”

The process analysed in this paper regards the direct use of 3D reality-based models in the FEA process and the error in the analysis compared to the experimental test inlaboratory, so no information of the subsurface have been acquired.

“A considerable amount of research has been done combining digital image correlation which can give very accurate in-plane and out of plane strain fields and finite element analysis. What are the relative advantages and disadvantages of the two techniques?”

We have not considered the digital image correlation method in this work. We have added it for future work

“A considerable amount of research has been done on generating FEM meshes from computed tomography point clouds. This was not mentioned in the paper nor was how these methodologies relate to developing FEM meshes from photogrammetry and laser scanning point clouds. It seems that some discussion of this would improve paper.”

We do not have used tomography here because the intention was to identify the error between the esperimental results and the FEA on a reality-based model. Since the FEA is usually done on CAD models without the use of tomography, the comparison was only between 3D models.

“It seems like there is too much detail on some well-known areas such as how strain gauges work and mesh elements. If this is important for the audience, it seems a reference would be better.”

We have added references and checked the English

Reviewer 2 Report

Comments and Suggestions for Authors

Add more bibliographic references.

Test the approach with different scanner to avoid problems about metal surfaces.

Comments on the Quality of English Language

The quality of English is good.

The text reads fluently and is easy to understand.

Author Response

We thank the reviewer for their suggestions. We have answered each review below.

“Add more bibliographic references.

Test the approach with different scanner to avoid problems about metal surfaces”

We have added references and will do in future test with different scanners. We added a sentence in the conclusions.